# UXO-AID: A New UXO Classification Application Based on Augmented Reality to Assist Deminers

**Qabas A. Hameed, Harith A. Hussein, Mohamed A. Ahmed *, Mahmood M. Salih, Reem D. Ismael and Mohammed Basim Omar**

Department of Computer Science, College of Computer Science and Mathematics, Tikrit University, Tikrit 34001, Iraq
* Correspondence: mohamed.aktham@tu.edu.iq

**Abstract:** Unexploded ordnance (UXO) is a worldwide problem and a long-term hazard because of its ability to harm humanity by remaining active and destructive decades after a conflict has concluded. In addition, the current UXO clearance methods mainly involve manual clearance and depend on the deminer's experience. However, this approach has a high misclassification rate, which increases the likelihood of an explosion ending the deminer's life. This study proposes a new approach to identifying the UXO based on augmented reality technology. The methodology is presented based on two phases. Firstly, a new dataset of UXO samples is created by printing 3D samples and building a 3D model of the object data file with accurate data for 3D printed samples. Secondly, the development of the UXO-AID mobile application prototype, which is based on augmented reality technology, is provided. The proposed prototype was evaluated and tested with different methods. The prototype's performance was measured at different light intensities and distances for testing. The testing results revealed that the application could successfully perform in excellent and moderate lighting with a distance of 10 to 30 cm. As for recognition accuracy, the overall recognition success rate of reached 82.5%, as the disparity in the number of features of each object affected the accuracy of object recognition. Additionally, the application's ability to support deminers was assessed through a usability questionnaire submitted by 20 deminers. The questionnaire was based on three factors: satisfaction, effectiveness, and efficiency. The proposed UXO-AID mobile application prototype supports deminers to classify the UXO accurately and in real time, reducing the cognitive load of complex tasks. UXO-AID is simple to use, requires no prior training, and takes advantage of the wide availability of mobile devices.

**Keywords:** unexploded ordnance; augmented reality; mobile application; intelligent systems



## 1. Introduction

Unexploded ordnance (UXO) is a global problem and an ongoing threat due to the possibility of its remaining active and potentially explosive even decades after a conflict has ended, as reported by L. Safatly et al. [1]. The most significant problem posed by UXO is that it endangers civilian lives. Furthermore, the risk of explosives impacts the lives of troops and explosives specialists, as well as the country's development. Unexploded ordnance is increasing in continued wars, such as the Russian–Ukrainian conflict. More than 54,000 unexploded ordnances were located and destroyed in the first month of the Ukraine–Russia crisis [2], Ukraine being one of the countries most highly contaminated by unexploded ordnance. The Landmine and Cluster Munition Monitor published its 23 annual reports in 2021 [3], which stated that 2020 was the sixth consecutive year that recorded a high number of casualties due to increased conflict and unexploded ordnance pollution since 2015. The report also said that 80% of the deaths were civilians (4437), with children constituting at least half of the civilian casualties (1872) while the remaining 20% consisted of military casualties (1105). Furthermore, as stated by the Geneva International

Centre for Humanitarian Demining (GICHD), there are over 60 countries contaminated with (UXO) [4].

Demining or explosive removal are the sole options for eliminating risks of unexploded ordnance, despite the dangers, time-consuming, and cost. Demining approaches, in general, are classified into three types: mechanical clearance, robotic clearance, and manual clearance, according to R. Achkar et al. [5]. Animals and machine clearance methods have been increasingly used in demining operations. However, most UXO and ERW are still removed using the manual clearance method, as reported by M. A. V. Habib [6]. Therefore, it is impossible to dispense with the intervention of specialist human operators in the removal of mines and explosives, despite the threat of wounding or loss of life. According to reports [3] from 2017 to 2020, over 250 human operators were killed and injured, although the number may be higher, since the report's coverage was limited to specific regions.

Recently, researchers developed artificial intelligence-based strategies to assist specialists and human operators in detecting explosives. K. Tbarki et al. [7] used one-class classification to detect and locate whether the buried object was UXO or clutter. The GPR data were used as input to the classifier to classify whether the detected object was a UXO. The authors evaluated the proposed method by conducting a comparison study with other methods. Similarly, K. Tbarki et al. [8] used a support vector machine (SVM) for landmine detection. The GPR data were utilised as input to the (SVM). The authors measured the performance of their proposed method with various techniques, such as receiver operating characteristic (ROC) and running time. The results indicated that the method was successful in landmine detection. The online dictionary learning technique was developed by F. Giovanneschi et al. [9] to form the received GPR data into sparse representations (SR) to improve the feature extraction process for successful landmine detection. This method takes advantage of the fact that much of the training data are likely correlated. The authors conducted a comparison study with three online algorithms to evaluate the proposed method. F. Lombardi et al. [10] presented adaptable demining equipment based on GPR sensors and employed convolutional neural network (CNN) to process the GPR data to detect the buried UXO. The results of the experiments showed excellent distinguishing accuracy between UXO and clutter objects.

Metal Mapper is another sensor used to detect buried objects. J. B. Sigman et al. [11] developed an automatic detection approach based on the Metal Mapper sensor coupled with the supervised learning technique naïve Bayes classifier. The proposed method can automatically detect the UXO without requiring user intervention, reducing cost and time. Furthermore, the minimum connected component (MCC) approach based on the mathematical concept of graph theory was presented in V. Ramasamy et al. [12] to detect buried UXO. The method explored the 2D image output from the GPR sensor. The proposed method demonstrated its effectiveness in feature extraction in the training datasets. Another form of hardware used is a metal-detector handheld device designed to identify suspicious objects containing metallic components. L. Safatly et al. [1] proposed a landmine recognition and classification method. A robotic system with a metal detector was designed to build the dataset. The authors used several machine learning algorithms, such as boosting bagging and CNN, to evaluate the system's precision in discrimination between UXO and clutter objects. Computer vision was also explored to detect and classify the landmines. R. Achkar et al. [5,13] proposed a robot to detect the landmines and identify the type and model by employing a neural network. A. Lebbad et al. [14] suggested a system based on computer vision focused on landmine classification issues by developing an image-based technique that utilised neural networks trained on a self-built and limited dataset.

Most researchers focused on detecting the explosives or distinguishing between UXO and clutter, ignoring the identification of UXO and its properties. Identifying UXO is critical to assisting operators in the minefield in avoiding mistakes and thus saving their lives by providing valuable information regarding the recognised object. Therefore, developing an application capable of identifying explosives and providing the minefield operator with information about the recognised object is required.

Augmented reality (AR) is a promising technology for supporting users in performing complex tasks and activities due to its ability to incorporate digital information with users' real-world perceptions, as shown by E. Marino et al. [15]. AR is becoming increasingly widespread in supporting operators in the workplace by reducing human mistakes and reducing reliance on operator memory, according to D. Ariansyah et al. [16]. AR applications have been developed and effectively deployed in a variety of fields, such as education, as in M. N. I. Opu et al. [17], heritage visualisation, as in G. Trichopoulos et al. [18], and training, as in H. Xue et al. [19].

Various AR applications were proven to be effective, valid, practical, and reliable approaches to risk identification, safety training, and inspection in the study by X. Li et al. [20]. AR can recognise unsafe settings and produce potential scenarios and visualisations using conventional safety training methods, as shown by K. Kim et al. [21]. AR is also employed as a platform for presenting immersive visualisations of fall-related risks on construction sites, as reported by R. Eiris Pereira et al. [22]. Furthermore, the capabilities of AR in presenting various visual information in real time have proven to be an advantage in emergency management and a better alternative than traditional methods, such as maps, according to Y. Zhu et al. [23].

In the literature, two studies based on AR are presented with relevance for the UXO field. T. Maurer et al. [24] developed a prototype that integrates AR with embedded training abilities into handheld detectors, improving the training process by enhancing the operator's visualisation with AR in order to examine the locations of a previously scanned area. Golden West Design Lab has implemented a marker-based AR system (AROLS) in Vietnam to train the local minefield operators. A set of markers were designed that start the AR process, as reported by A. D. J. T. J. o. C. W. D. Tan [25]. Both studies provided an AR application for training minefield operators that was only intended to be used indoors. Furthermore, these applications were unable to identify UXO or offer relevant information, and therefore did not support UXO deminers during the clearance process in real time. To the best of the author's knowledge, there is no application designed to identify UXO types using the AR technique.

This research proposes a new augmented-reality-based approach for identifying UXO types in real-time, enhancing field operators' productivity, and assisting in the disposal process. The main contributions of this paper are as follows.

1. The proposed application presents a unique, innovative, and inexpensive UXO classification method through AR technology.
2. The proposed application provides information in real time related to the detected object.
3. It can reduce the risk imposed on deminers during UXO clearance operations by displaying visual information illustrating the type and the components of the UXO.
4. An evaluation study in a different setting and a questionnaire were conducted to measure the performance and usability of our proposed application.

This paper is organised as follows: Section 2 defines the research background. Section 3 presents a detailed description of the methodology. The experimental process is described in Section 4. Section 5 defines the evaluation of the application through a usability test. Finally, the conclusion and future work are presented in the section.

## 2. Research Background

This section discusses UXO activities such as risk management, clearance methods, and UXO types. Finally, we review the AR-based assistance applications with an object recognition approach.

### 2.1. UXO Risk Assessment and Clearance Methods

UXO risk consists of two main kinds. The first risk is related to the hazard of explosion. Humans are injured, dismembered, or killed when exposed to explosive UXO. The second risk is the damage caused to the soil and the environment due to the leakage of chemical material into soil and water. Furthermore, the types of UXO located in the sites differ

greatly based on the kinds of wars, conflicts, and military training that happened at the location; UXO can range from small ordnance to extremely large. Furthermore, every UXO contains different charges of explosive materials. Before any operations occur, every governmental and humanitarian organisation assesses the possibility of the existence of UXO. UXO risk assessment can be defined as analysing and evaluating the likelihood of detecting UXO, as asserted by J. MacDonald et al. [26].

The UXO risk assessment operations consist of four main stages (see Figure 1): preliminary assessment, comprehensive assessment, UXO mitigation plan, and UXO collecting, followed by detonation, as each step depends on the primary outcome. In the primary assessment stage, a simple scanning inspection is conducted by checking whether there is a history of military activity, such as training and weapons testing at the site, or whether the site location was a part of a war conflict. Finally, an inspection is carried out regarding any reports of UXO detection regarding that site.

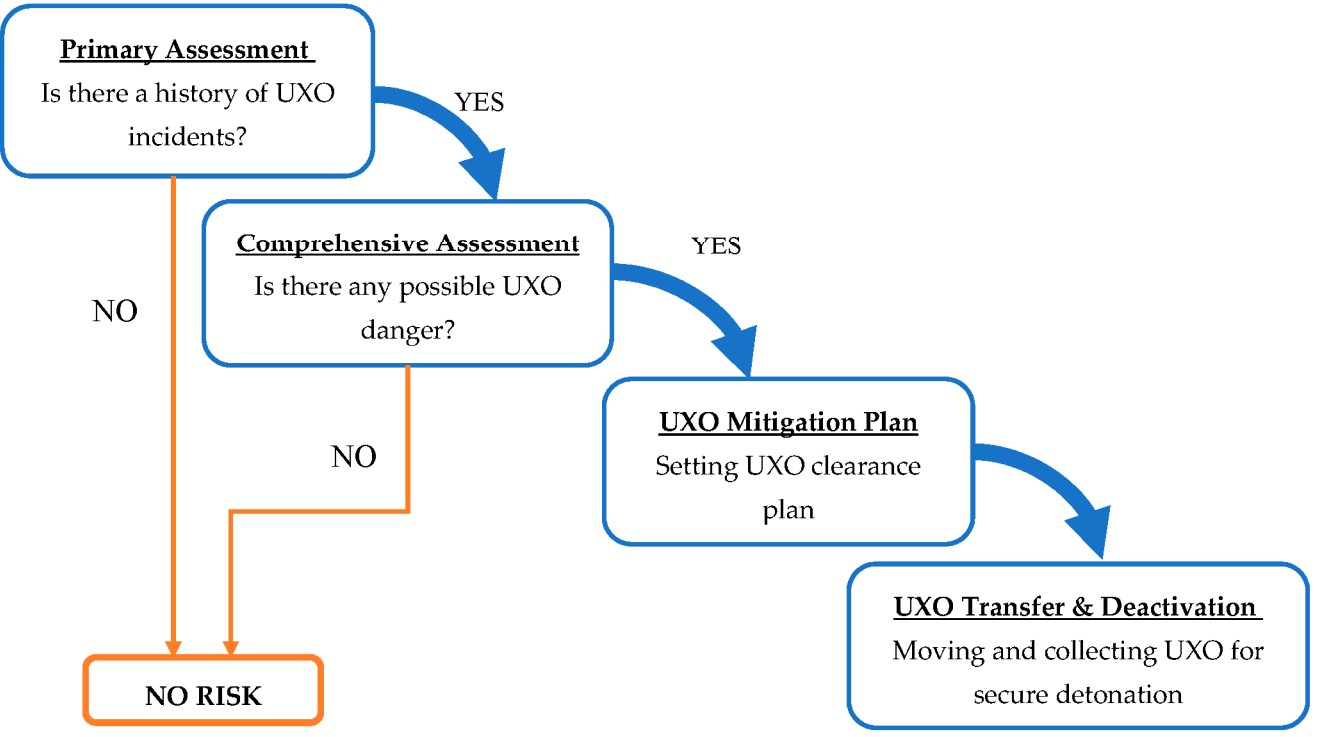

**Figure 1.** UXO risk-assessment stages.

The outcome of this stage is to determine whether there is a need for a comprehensive assessment. If a thorough evaluation is required, a detailed scanning and in-depth groundwork are conducted to determine if there is a high potential risk of UXO on the site. When the outcome of this stage indicates that the site is probably contaminated, a UXO clearance plan is devised that includes UXO detection and classification. After detecting and classifying the UXO, the deminers collect and transfer the various UXO to a secure and isolated place. The authorities set a date to destroy the collected UXO and announce the date in a public announcement to protect public safety and avoid possible accidental explosions.

According to H. Kasban et al. [27], UXO detection has different methods, including electromagnetic, mechanical, and biological detection. Metal detectors, GPR sensors, and IR techniques are electromagnetic detecting methods. Mechanical detection methods include using mechanical equipment, such as a vehicle, that moves the UXO from its location, which causes the explosion of UXO. Biological detection approaches include dogs, bees, rodents, bacteria, and plants. The effectiveness of each method is measured according to the nature of the contaminated soil and the features of the UXO.

### 2.2. AR with Object Recognition

Augmented reality (AR) and object recognition are among the most advanced technologies that have seized the attention of researchers, and the integration of these two technologies enables a unique approach to solving diverse problems. Manufacturing and operator assistance in performing tasks and training have benefited most from integrating these two technologies due to Industry 4.0., because AR is an appropriate method for assisting operators during the performance of complex activities by offering guidance to make their performance more efficient, which helps to reduce human errors caused by distraction or inexperience; in such systems, object recognition was accomplished using Mask R-CNN, as reported by K.-B. Park et al. [28] and YOLO, as reported by H. Bahri et al. [29] R-CNN Z.-H. Lai et al. [30].

L. Zheng et al. [31] introduced an AR system to support cable assembling and inspection operations. The system used deep learning to detect and identify cable brackets and CNN to capture the labels on the cables simultaneously; AR presents visual guidance in real time. Moreover, AR combined with object recognition are evolving as a favoured approach for training and technical support because of its capability to provide an intuitive method to convey detailed information, as shown by C. Piciarelli et al. [32] and B. Zhou et al. [33].

Another domain that utilizes AR with object recognition is that of driving assistance systems. In R. Anderson et al. [34], YOLO and Viola-Jones were two detection methods used to detect roads and different obstacles, while the AR part displayed holograms to increase the driver's awareness, hence improving driver safety and helping to reduce fatal accidents. Another domain that utilizes AR with object recognition is the driving assistance systems. L. Abdi et al. [35] presented an in-vehicle assistance system that consists of three parts, a wireless network, an AR virtual information projected on the car windshield, and CNN for object detection.

In the learning and education domain, AR and object recognition integration is a promising medium that allows students to understand information quickly. Moreover, the learning process itself becomes more engaging and pleasant. B. Huynh et al. [36] presented an in-site language learning framework. The proposed framework used SSD as an object recognition method and an AR component that attached a virtual label to the detected object in a different language. In E. F. Rivera et al. [37], AR application was developed as a learning and training tool for studying automotive engineering. The objective was to demonstrate to the students the various driving conditions of the contrasting functions and status of the power splitting device. The application was designed with a Vuforia (SDK) object scanner and CAD software to create 3D models.

### 3. Methodology

This study aimed to design and develop an augmented reality (AR) application to support deminers by classifying the type of UXO and providing virtual information to assist the deminers in making the appropriate decisions, thus reducing the risks.

The application identifies UXO and provides contextual information to the operators. This information can take the form of different mediums, such as text, images, 3D, animation, and videos. To design and build the proposed application for aiding the demining operation, we needed to collect all the information related to the deminer workflow and the procedure used in classifying the UXO. Hence, we conducted unstructured interviews with deminers to understand the limitations and problems encountered during their operations in collecting information. In addition, we used the suggestions as a guideline to build the application. As for the technological aspect, we chose AR techniques to develop the virtual assistance content. AR technology met all the requirements for showing all the required information effectively and in real time. With the benefit of AR techniques, the deminers viewed the virtual information in the real world. Finally, we evaluated the application of a qualitative analysis-based questionnaire presented to 20 minefield operators to measure the usability of the proposed prototype. Figure 2 depicts the methodology used in this study.

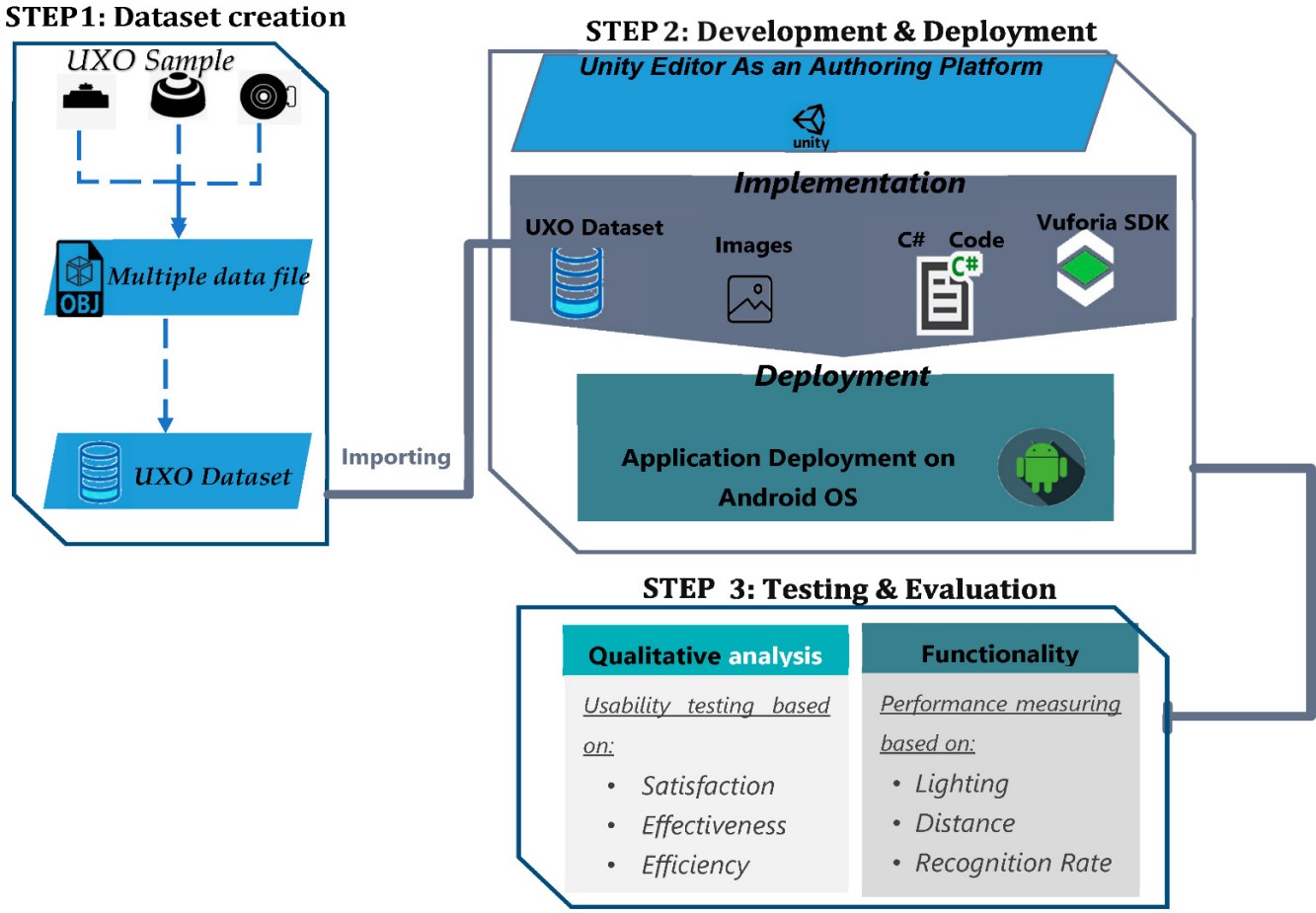

**Figure 2.** The methodology of the study.

### 3.1. Data Gathering

Data gathering involved conducting interviews to collect valuable information associated with the study. The interview procedure is an appropriate and productive method for obtaining knowledge and gathering information on a specific issue. Therefore, we interviewed experts in UXO detection and clearance. Consequently, an unstructured interview was conducted with two experts working at the Explosive Control Directorate (EOD)—Salaheddin department. They shared their knowledge and experience about UXO clearance and classification and emphasised the significance of various aspects of the UXO procedures. In addition, they provided their thoughts and suggestions to help decide on the most suitable approach to designing the system and the appropriate aid required to support their operations in the field. According to the experts' feedback, the procedure of identifying and classifying UXO is outlined in a paper manual comprising simple illustrations and labels assigned to different components of UXO. After completing the interview, photos of UXO samples were acquired see Figure 3. However, working with actual samples of UXO was not possible because the detonator triggers were still active, which imposed safety issues.

### 3.2. Dataset Collection

Datasets are fundamental to fostering AI development, giving results scope, robustness, and confidence. This study's dataset collection process included two phases: a 3Dprinting sample and a 3D mobile-based model.

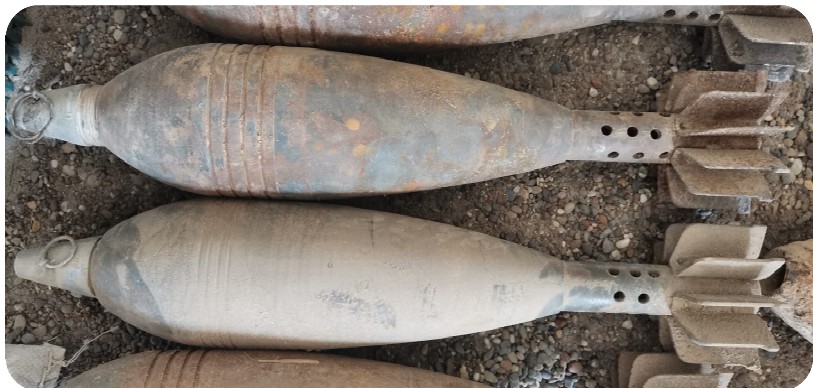

**Figure 3.** UXO sample.

### 3.2.1. 3D Printing Sample

Considering the danger of working with the active UXO and the authors' lack of experience manually handling the UXO, 3D printing technology was chosen to create the UXO samples. 3D printing has evolved swiftly in recent years, making it possible for ideas to transfer to reality quickly. Furthermore, light and elastic printing materials, such as ABS or PLA, are inexpensive and contribute to reducing production. For this purpose, the Creality 3D CR-10S Pro V2 printer with FDM 3D printing technology using polylactic acid (PLA) material was used to print the UXO samples. Four types of UXO were chosen as the reference objects, namely, MPN-2, VS-MK, VS-50, and 45mm mortars, as shown in Figure 4. We chose these UXO samples based on the advice of the deminers we met during the data gathering process, since they are among the most prevalent UXO found in minefields. In addition, the four types selected differed in size and shape, helping in the evaluation of the proposed prototype's performance. Figure 5 illustrates the samples of unexploded ordnance chosen in this study.

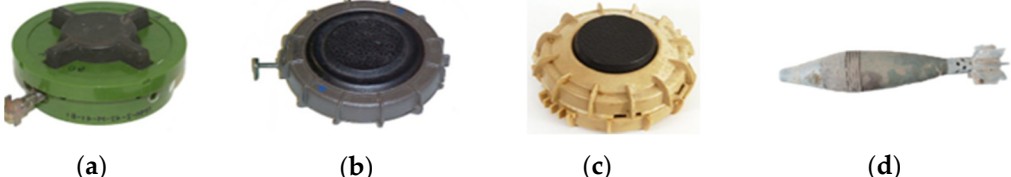

    (**a**)                (**b**)                (**c**)                (**d**)

**Figure 4.** UXO samples: (**a**) PMN-2 soviet mine; (**b**) VS-MK soviet mine; (**c**) VS-50 Italian mine; (d) 45 mm mortars.

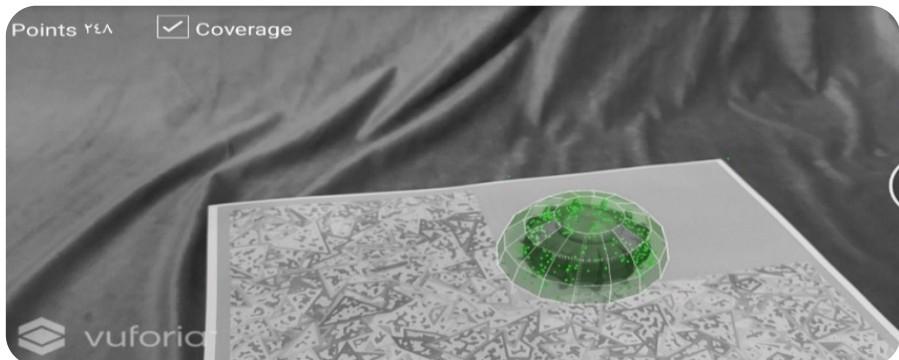

**Figure 5.** UXO scan 3d.

Three-dimensional printing processes involve two stages, namely modelling and printing. We designed the UXO model in the modelling phase, utilising a computer-aided design (CAD) software package. The object model was saved in stereolithography (STL)

format when finished. The item was built in the second phase of 3D printing (i.e., the printing phase). To begin printing the specified UXO, a 3D (STL) format file was uploaded to a 3D printing machine. The instructions in the corresponding file will be used by the 3D printer to determine where and how the material is deposited.

### 3.2.2. 3D Mobile-Based Model

After creating the UXO samples, the dataset can be built using the android application Vuforia Object Scanner, which operates as a scanner by using the mobile camera to make a 3D model. The scanning process creates the object data file (OD) that includes accurate data for targeting objects in the Target Manager. A series of prerequisites must be completed for a successful scanning operation, such as acquiring and printing the Object Scanning Target from the Vuforia website. Furthermore, the environment must be free from background noise with moderate and uniform lighting. Therefore, a grey background was used to minimise the noise. In addition, several scanning processes were conducted to obtain the best result. Once the scanning process was completed, the (OD) file was saved for future import into the Unity software. Figure 5 shows the scanning process.

### 3.3. UXO-AID Overview

The core of the presented mobile application is illustrated in Figure 6. The application layout is simple to provide a user-friendly interface that allowed quick perception. Firstly, feature extraction is applied to the object of interest from diverse viewpoints and distances, and saved to the application's database in a 3D target model. UXO-AID then acquires a frame from the mobile phone camera, extracts the features, and performs a matching process between the extracted features and the features of the 3D model saved in the database. In the event of a successful matching process, the object is successfully detected and recognised. Then the virtual graphics and text are displayed. The virtual content guides the minefield operator on the location of the denoting trigger in the UXO. Moreover, additional information is provided about the model, type, and how the UXO is actuated.

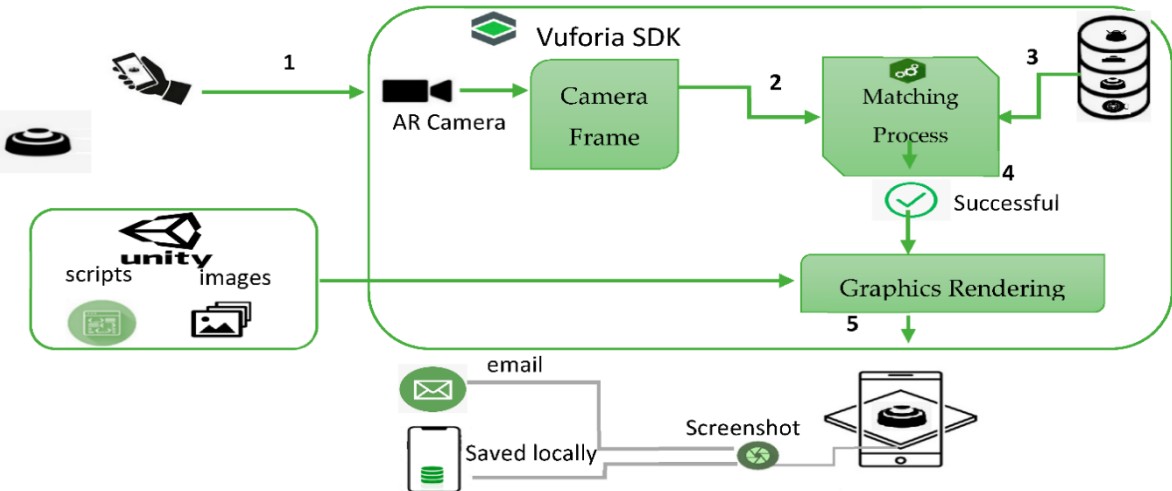

**Figure 6.** Application overview.

In addition, the operational flow of the UXO recognition process using the smartphone's camera and the sequence of actions taken by the user for successful recognition are shown in Figure 7. The steps illustrated in the flowchart are:

1.  Open camera: This is the first step, where the deminer opens the mobile phone's camera.
2.  Scan object: After opening the camera, the deminer must point the camera to identify the UXO type.

3. Object detection: The camera recognises the UXO's appearance, which matches the target model already stored in the database. If the detected object is identified, the operation proceeds to the next step; if not, the operation returns to the scanning stage.
4. Render virtual information: Once the UXO type is recognised, the deminer sees all the relevant information of the UXO type.
5. Screenshot: After viewing the virtual information, the deminer can take a screenshot of the augmented view, which is saved on the mobile phone's memory.

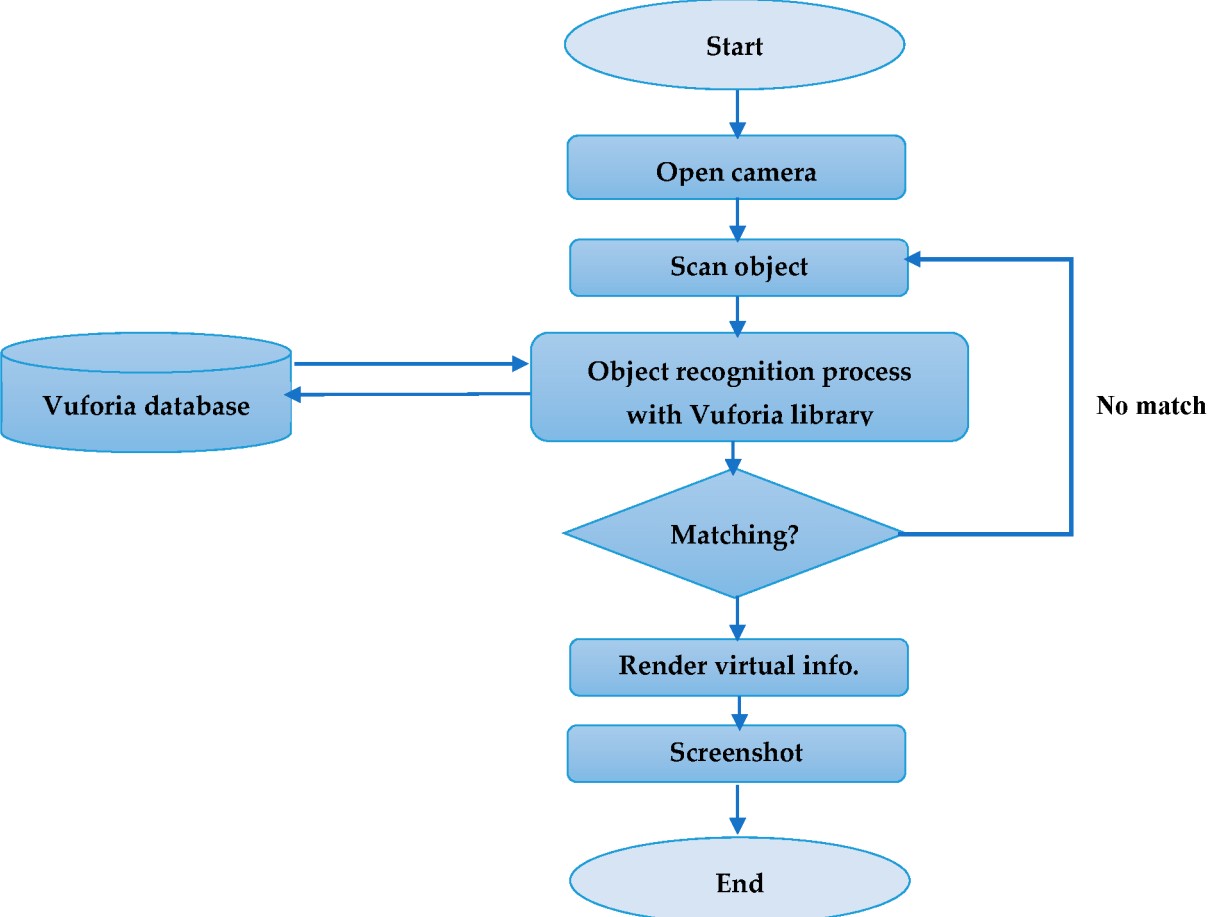

**Figure 7.** Workflow of the object recognition.

*3.4. Software Tools*

3.4.1. Vuforia SDK

Vuforia contains a set of diverse functionalities that support several types of recognition, such as image, text, and object recognition. Moreover, it is a widely used platform for creating augmented reality applications due to its compatibility with many mobile devices, such as tablets and mobile phones, and supports Android, iOS, and WUP. Vuforia can recognise 3D objects in real time using computer vision technology, and allows for direct interaction between the user and the real world. The mobile screen is a portal that combines the real-world scene with virtual content. In addition, it provides services related to recognition in different modes, such as online and offline. Cloud recognition works online, allowing developers to store and handle targets online. The supported types of targets include 2d images and planer. Meanwhile, the offline mode enables developers to recognise 2d images, planer, and 3d objects, and hosts the different targets locally on the device, which removes the need for internet service to use the application.

Furthermore, Vuforia provides the natural feature tracking (NFT) tracking technique. NFT is a model-based or image-based approach that recognises and tracks the previously

extracted natural features from the target object. The goal is to facilitate the process of feature pattern detection. The Vuforia SDK employs NFT (natural feature tracking) that contains a natural feature tracking method, i.e., SIFT (scale invariant feature transform). SIFT is used to detect the feature points of the object and calculate the object's scale by mapping the coordinate values.

### 3.4.2. Unity 3D Engine

Unity is a cross-platform 2D–3D game engine that supports 2D and 3D graphics and C# scripting, in addition to animations designed by the Unity 3D engine. It is a powerful engine used to build AR and VR applications, providing a basic level of human-machine interaction using AR development tools. Furthermore, Unity is compatible with Vuforia SDK, allowing for the building of AR applications capable of recognising and tracking 3D objects.

### 3.5. UXO-AID Implementation

For the implementation of the application, Android was selected as the target mobile operating system (OS) since it is considered one of the most commonly used OSs in the world. As this research aimed to use AR technology to assist the deminers during their tasks, adaptation to additional platforms was unnecessary. Then a comparison of the current AR SDK was conducted to choose a suitable tool. Different AR SDKs and libraries are used for creating AR applications, such as Vuforia, ARCore, ARToolkit, easyAR, Wikitude, and Kudan. However, when comparing these AR SDKs, Vuforia was found to offer better 3D object recognition and tracking. As a result, Vuforia was the suitable SDK for implementing the application. The prototype was created using Unity (version 2019.4.35f1 (64-bit)) with a Vuforia plug-in. After selecting Vuforia SDK, the UXO model dataset was created and imported, as mentioned in Section 3.2. The main criterion for successful object recognition is the acquisition of as many feature points as possible to facilitate the Vuforia engine to recognise the object easily. Then the application user interface was created using Unity's GameObject-based interface design tool and immediate mode GUI (IMGUI) coding API. The gameObject-based feature allows the addition of various user interface components, such as canvas, texts, and buttons, simply by placing the details on the screen. The application has two different user interfaces: the home interface and AR mode. The home interface is displayed when the user starts the application. Figure 8a shows the home interface with two buttons; open camera and exit buttons. Figure 8b shows the second interface, AR mode, which opens the camera view of the mobile's camera feature, allowing the users to scan the target object. In addition, the interface has Screen-shot, Back, and Exit buttons. After completing the design of the application user interfaces, scripting was added to expand the application's features and enable the required capabilities that allow the application to be dynamic. Therefore, three C# scripts were written. The first script is linked with a button to screenshot the displayed virtual content associated with the UXO; an open-source package (UnityNativeGallery) was imported to use the feature of saving the screenshot on the mobile gallery. The second script is an autofocus script associated with the AR camera. Lastly, the third script is linked with the Exit buttons to close the application and the Back button to return to the main menu.

### 3.6. Evaluation of UXO-AID

This section describes the assessment of the performance of the proposed prototype. To accomplish this assessment, firstly, a preliminary experiment was conducted to determine the application's applicability. The testing process focused on distance in centimetres, lighting, and object recognition accuracy. The experiment aimed to determine if the application could run correctly when the object was scanned from different distances in conjunction with both good lighting and low lighting conditions. Furthermore, to measure the frequency of recognition, each UXO was scanned individually 20 times using a mobile phone camera. Figure 9 shows the procedure to measure the object recognition accuracy.

Next, the accuracy was computed using Equation (1), while the error rate was calculated using Equation (2), as listed below:

$$\text{Accuracy \%} = \frac{\text{UXO recognised}}{\text{no. of attempts}} \times 100 \tag{1}$$

$$\text{Error not recognised} = \frac{\text{UXO not recognised}}{\text{no. of attempts}} \times 100 \tag{2}$$

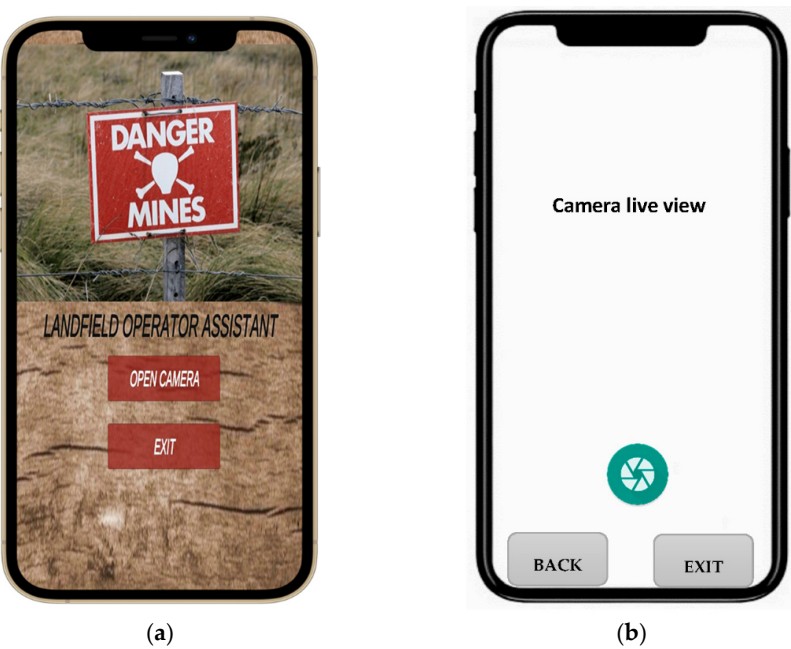

**(a)**       **(b)**

**Figure 8.** Application user interfaces: (**a**) home interface; (**b**) AR mode interface.

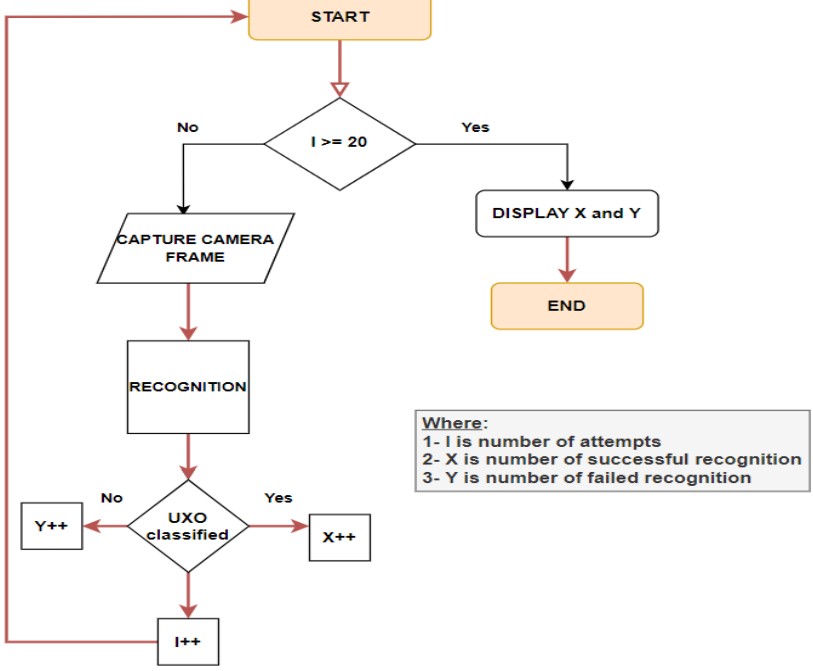

**Figure 9.** Flowchart of UXO testing.

Secondly, usability testing was performed to determine the level of applicability and usability of the AR application for participants, and to examine whether the AR application could support the deminer operations and to what extent the displayed virtual information regarding the recognised UXO assisted the operator. A questionnaire based on usability was completed and submitted by 20 deminers. According to N. Bevan [38], usability is defined as how a user may utilise software to accomplish an objective with satisfaction, efficiency, and effectiveness for a specific usage context. Therefore, the questionnaire was broken down into three significant factors to be considered in assessing the AR application: satisfaction, effectiveness, and efficiency, as described in Table 1. Each factor was measured using items derived from different studies [39–41]. The assessed items for each factor are also detailed in Table 1. After collecting the responses from the deminers, descriptive statistics were applied to analyse the questionnaire. Hence, maximum (max), minimum (min), mean, standard deviation (std), tabled T and calculated T were implemented. Figure 10. Demonstrate the evaluation techniques used in this study.

**Table 1.** Implemented factors with their associated measured item.

| Factor | Definition | Measured Item |
|---|---|---|
| Satisfaction | Satisfaction is influenced by likeability, functional appropriateness, and simplicity of use [38] | • The AR application is useful<br>• The user interface is simple to navigate and easy to learn<br>• The app will be used frequently<br>• In general, it is easy to use the app<br>• I believe I could become productive quickly using this system |
| Effectiveness | Effectiveness means an objective can be measured in accuracy, completeness, and output precision [42]. | • The response of the AR application in real-time<br>• The AR application does not have any issues at the time of execution<br>• The digital information properly presented<br>• The virtual information was effective in helping me complete the tasks. |
| Efficiency | Efficiency is the resources utilised, such as time to finish a particular task, human labour, and cost [42]. | • The classification operation time decreased through using the application<br>• The application helps reduce the hazard imposed by the clearance operation<br>• The application helps reduce the mental demand to identify the UXO<br>• This application has all the functions and capabilities expected it to have |

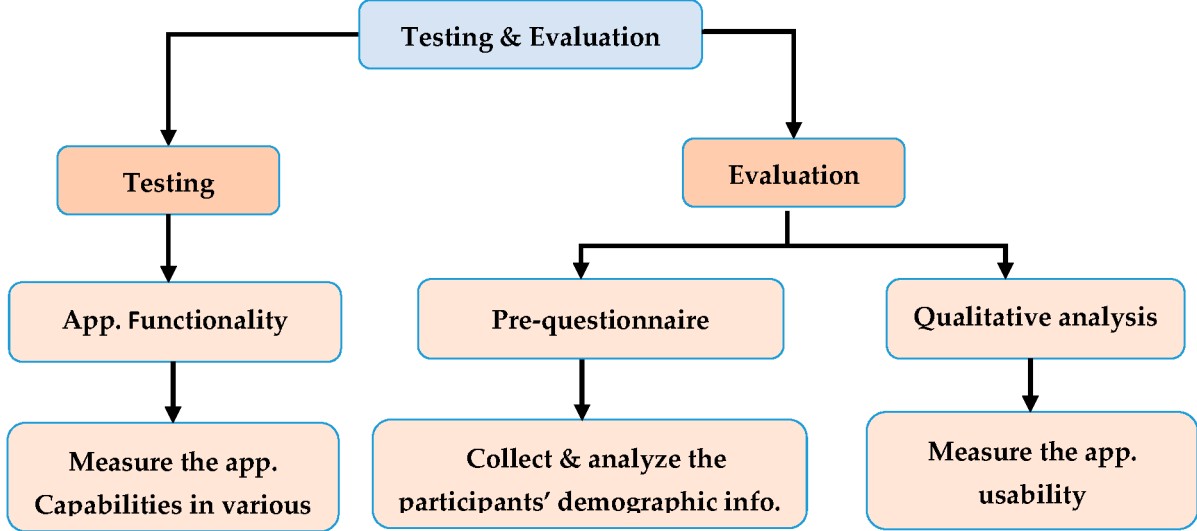

**Figure 10.** Testing and evaluation methods used in the study.

## 4. Experimental Results

### 4.1. Preliminary Tests

A series of preliminary experiments were carried out to measure the application functionality and to assess the application's applicability regarding distance and lighting settings. Table 2 depicts the testing result from six different distance points with varying light settings. The results indicate that the objects were easily recognised from all the distance points except when the distance was about 40 cm. In addition, scanning in light intensity of less than 12 LUX made UXO objects undetectable and unrecognisable, preventing them from being identified. Figure 11 shows the result of testing using the VS-50 sample. Figure 11a displays the camera view in landscape mode, and the VS-50 recognised within a distance equal to 20 cm. Figure 11b displays the camera view in landscape mode, and the VS-50 identified within a distance equal to 30 cm. On the other hand, the images of Figure 11c,d display the camera view in portrait mode, and the VS-50 recognised within distances equal to 30 and 35 cm.

**Table 2.** Various distance and lighting test samples results.

| Distance (cm)　　　Lighting Intensity | 10 | 15 | 20 | 25 | 30 | 40 |
|---|---|---|---|---|---|---|
| 2445 (LUX) | YES | YES | YES | YES | YES | NO |
| 2197 (LUX) | YES | YES | YES | YES | YES | NO |
| 1328 (LUX) | YES | YES | YES | YES | YES | NO |
| 843 (LUX) | YES | YES | YES | YES | YES | NO |
| 431 (LUX) | YES | YES | YES | YES | NO | NO |
| 380 (LUX) | YES | YES | YES | YES | NO | NO |
| 177 (LUX) | YES | YES | YES | YES | NO | NO |
| 35 (LUX) | YES | YES | YES | NO | NO | NO |
| Less than 12 (LUX) | NO | NO | NO | NO | NO | NO |

As for object recognition accuracy, it was found that the small number of feature points caused the recognition process to fail. In the case of VS-50 and VS-MK mines, the recognition rate was above 90%, because these two types had a sufficient number of feature points. On the other hand, the recognition rate decreased for the PMN-2 and 45 mm mortars due to insufficient feature points. Furthermore, the cylinder shape of the 45 mm mortars was another factor that caused the recognition process to fail. Finally, scanning the UXO in scenarios that contained light reflection and shadow also caused failure in recognition. Overall, the total accuracy of UXO recognition reached 82.5%. Table 3 shows the details of the testing results.

**Table 3.** Performance of UXO-AID.

| UXO Model | Recognised | Not Recognised | Accuracy |
|---|---|---|---|
| VS-50 | 20 | 0 | 100% |
| PMN-2 | 15 | 5 | 75% |
| VS-MK | 19 | 1 | 95% |
| 45 mm mortars | 12 | 8 | 60% |
| Overall Accuracy | | | 82.5% |

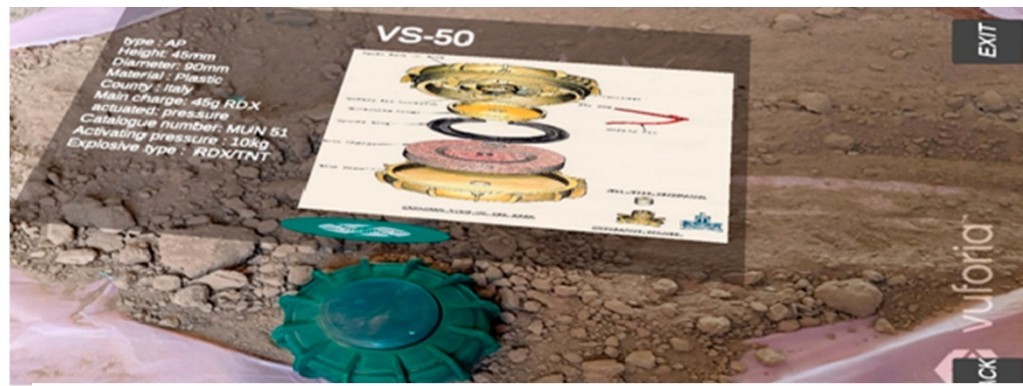

(**a**) VS-50 recognition in landscape mode with distance equals 20 cm

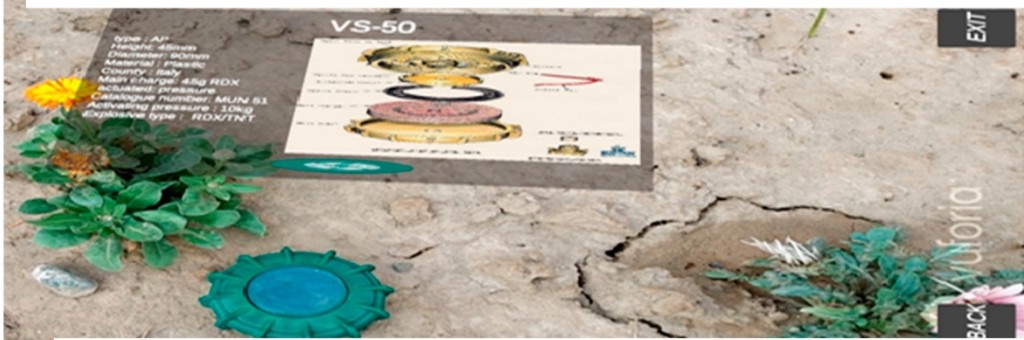

(**b**) VS-50 recognition in landscape mode with distance equals 30 cm

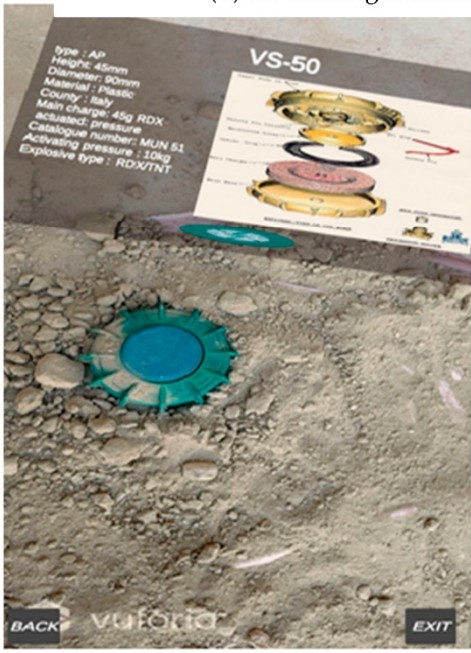

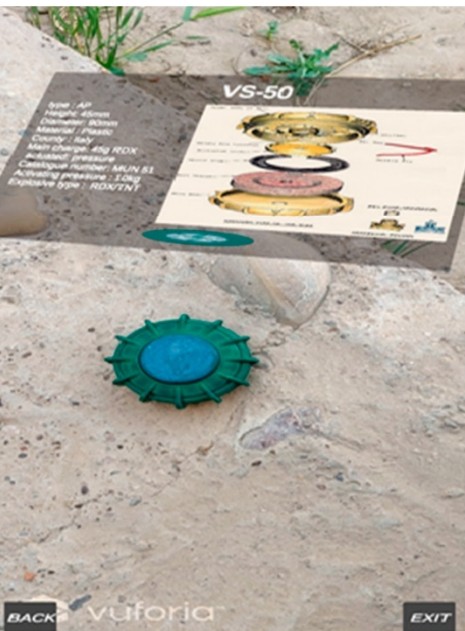

(**c**) VS-50 recognition in portrait
mode with distance equals 30 cm

(**d**) VS-50 recognition in portrait
mode with distance equals 35 cm

**Figure 11.** VS-50 recognition results in a different setting: (**a**,**b**) show the camera view in landscape mode with distances 20 and 30 cm; (**c**,**d**) show the camera view in portrait mode with distances of 30 and 35 cm.

*4.2. Usability Testing*

The complete usability testing consisted of two parts; in the first part, the participants were asked to fill out a pre-questionnaire based on their information. The purpose of the

pre-questionnaire was to collect relevant information from the participants. A group of 20 deminers (male 70% and female 30%) tested and evaluated the application. Table 4 outlines the participants' gathered demographic information. The pre-questionnaire focused on aspects concerning years of experience, estimation of the time taken to classify the UXO according to participants' answers, and whether the participants used any system to help them in the UXO classification process. Lastly, the participants' usage frequency of mobile devices was also considered.

**Table 4.** Demographics of the participants.

| Variables | Value | Frequency | Percentage % |
|---|---|---|---|
| Gender | Male | 14 | 70% |
| | Female | 6 | 30% |
| Education level | University | 16 | 70% |
| | Institution | 4 | 30% |
| | High school | 0 | 0% |
| Years' experience | 1–5 | 4 | 20% |
| | 5–10 | 7 | 35% |
| | 15–20 | 4 | 20% |
| | More than 20 | 5 | 25% |
| Number of training courses | One course | 4 | 20% |
| | Two courses | 4 | 20% |
| | Three courses | 4 | 20% |
| | Four courses | 5 | 25% |
| | More than four courses | 3 | 15% |
| Time to classify UXO | 1 h | 5 | 25% |
| | 2 h | 9 | 45% |
| | 3 h | 4 | 20% |
| | More than 3 h | 2 | 10% |
| Skills in using mobile apps | Excellent | 16 | 80% |
| | Good | 4 | 20% |
| | Medium | 0 | 0% |
| | Low | 0 | 0% |
| Did you use any system to assist you in the tasks | No | 20 | 100% |
| | Yes | 0 | 0% |

The analysis of the pre-questionnaire shows that all participants did not use any software during their operations. Furthermore, most participants had a university education (70%) and institutional education (30%). Regarding user experience, four participants (20%) had 1–5 years' experience, and seven participants (35%) reported that they had 5–10 years' experience, while another four participants (20%) had 10–20 years' experience. In contrast, five participants (35%) had more than 20 years of experience. Out of 20 participants, nine (45%) reported they could classify UXO within 2 h, and five participants (25%) said they could classify UXO within one hour. As for mobile skills usage, 16 participants (80%) reported that their skills in utilising mobile applications were excellent, whereas four participants (20%) stated that their skills in using the mobile application were good. Figure 12 illustrates the distribution of experience years and the required time to classify the UXO.

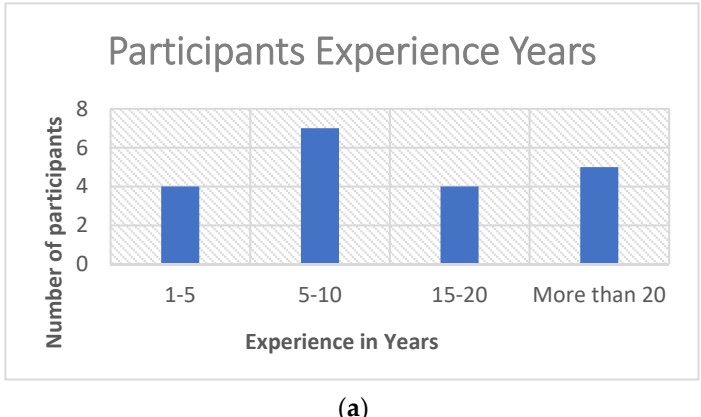
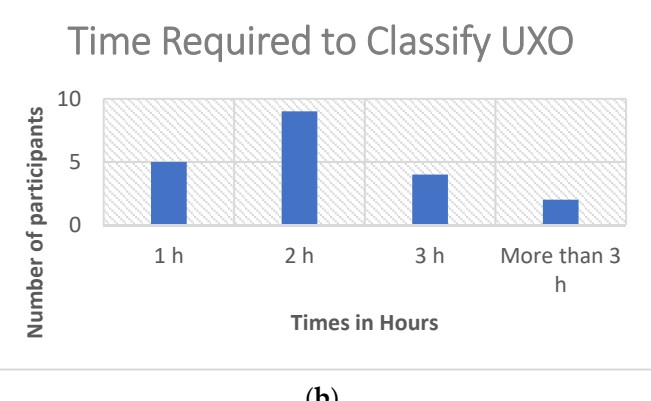

<p style="text-align:center">(<b>a</b>)　　　　　　　　　　　　　　　　(<b>b</b>)</p>

**Figure 12.** Distribution of pre-questionnaire regarding: (**a**) experience years.; (**b**) the required time to classify UXO.

In the second part of the evaluation, each participant who used the application answered the questionnaire. The participants were requested to convey their agreement level. The responses were rated on a five-point Likert scale, from strongly disagree (1) to strongly agree (5).

The result for each factor is discussed as follows.

- Satisfaction

This factor evaluated the level of user acceptance, impact, and ease of use of the application, and to measure this factor, five criteria were considered, as illustrated in Table 5. The majority of participants stated that the AR application was practical and could help the deminers in their operations to complete the tasks. Furthermore, many participants' answers indicated that they agreed or strongly agreed that the application was easy to use and did not require training to learn how to use the application, primarily due to the simplicity of the user interface and ease of navigation. This also influenced many participants who agreed that the application would be used frequently. However, another group chose to remain neutral, as they needed to use the application more before deciding to use it more regularly. In addition, many participants agreed that the application would increase their productivity in completing their tasks. At the same time, a small percentage did not agree with this statement. Table 6 shows the factor's overall statistical results, and implies that positive outcomes were acquired for the satisfaction factor obtained from the usage of AR.

**Table 5.** Descriptive statistics for the Satisfaction factor.

| Item No. | Strongly Agree | Agree | Neutral | Disagree | Strongly Disagree | Max | Min | Mean |
|---|---|---|---|---|---|---|---|---|
| 1 | 45% | 40% | 10% | 5% | - | 5 | 2 | 4.25 |
| 2 | 55% | 40% | - | 5% | - | 5 | 2 | 4.45 |
| 3 | 5% | 65% | 25% | 5% | - | 5 | 2 | 3.7 |
| 4 | 45% | 50% | - | - | - | 5 | 1 | 4.3 |
| 5 | 15% | 65% | 20% | 10% | - | 5 | 2 | 3.75 |

**Table 6.** Overall statistical analysis for the Satisfaction factor.

| Mean | Std. Deviation | Calculated Mean | Calculated T | Tabled T |
|---|---|---|---|---|
| 20.45 | 3.06 | 15 | 7.94 | 2.08 |

- Effectiveness

Effectiveness was another significant factor considered as a section of the assessment with four criteria. The results in Table 7 indicate that most of the participants agreed that the application response was in real time, and the participants did not wait for the virtual information to be displayed. However, a small group of participants remained neutral. Furthermore, most participants agreed or strongly agreed that the application operated without noticeable issues. Still, a few participants stated that the application encountered some problems during the execution. This was due to the old versions of mobile phones with low-resolution cameras. The participants showed good agreement that the virtual information displayed appropriately regarding the positions and orientations.

**Table 7.** Descriptive statistics for the Effectiveness factor.

| Item No. | Strongly Agree | Agree | Neutral | Disagree | Strongly Disagree | Max | Min | Mean |
|---|---|---|---|---|---|---|---|---|
| 1 | 35% | 50% | 15% | - | - | 5 | 3 | 4.2 |
| 2 | 20% | 45% | 30% | 5% | - | 5 | 2 | 3.8 |
| 3 | 25% | 50% | 25% | - | - | 5 | 3 | 4 |
| 4 | 20% | 35% | 45% | - | - | 5 | 3 | 3.75 |

Finally, a significant group of participants answered neutrally to the last statement. The participants proposed including a dynamic 3D model with the already displayed information for better assistance in completing the tasks. In general, the positive effectiveness results imply that end-users correctly and properly viewed the application outputs. Overall statical results on effectiveness are shown in Table 8.

**Table 8.** Overall statistical analysis for the effectiveness factor.

| Mean | Std. Deviation | Calculated Mean | Calculated T | Tabled T |
|---|---|---|---|---|
| 20.45 | 3.06 | 15 | 7.94 | 2.08 |

- Efficiency

The outcomes for this factor are presented in Table 9. Most participants agreed that the application would reduce the time used in classifying the UXO. Moreover, most participants strongly agreed with statement two that the application decreased the danger of explosion during the deminers' clearance operations. In addition, a few participants disagreed that the application helped reduce the mental load on the deminer. However, the majority agreed with the statement and indicated that the application reduced the cognitive efforts in classifying the UXO. A group of participants remained neutral with regard to suggestions to use smart glasses to allow the deminer to operate with both hands and incorporate remote collaboration in real-time. Table 10 depicts the overall statistical analysis for efficiency.

**Table 9.** Descriptive statistics for the Efficiency factor.

| Item No. | Strongly Agree | Agree | Neutral | Disagree | Strongly Disagree | Max | Min | Mean |
|---|---|---|---|---|---|---|---|---|
| 1 | 50% | 40% | 5% | 5% | - | 5 | 2 | 4.35 |
| 2 | 50% | 45% | - | 5% | - | 5 | 2 | 4.4 |
| 3 | 30% | 65% | - | - | 5% | 5 | 1 | 4.15 |
| 4 | - | 40% | 45% | 10% | 5% | 4 | 1 | 3.2 |

**Table 10.** Overall statistical analysis for the Efficiency factor.

| Mean | Std. Deviation | Calculated Mean | Calculated T | Tabled T |
|---|---|---|---|---|
| 16.1 | 2.79 | 12 | 6.574 | 2.08 |

The mean distribution of the three factors is shown in Figure 13, which demonstrates the positive results that were collected and indicates that each factor's mean value was higher than 3.5, verifying a high approval level.

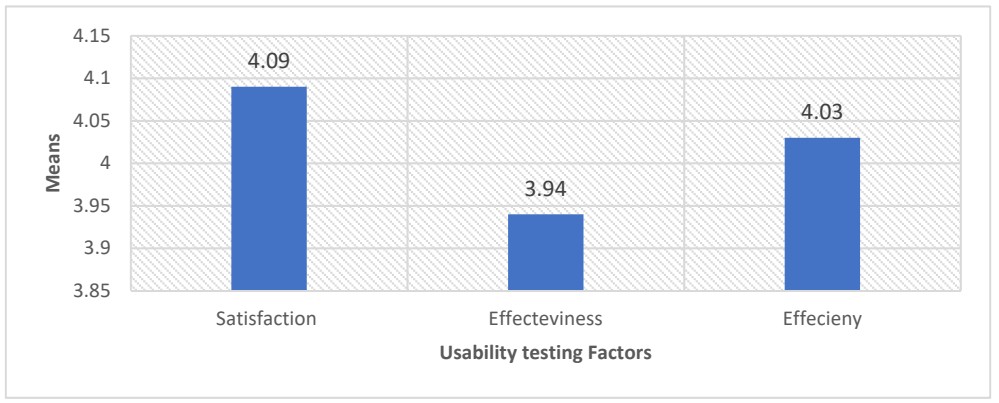

**Figure 13.** Mean distribution of each factor.

In addition, Figure 14 presents an overview of the mean of each measured item. From the chart, it is observed that participants agreed with the items Q1, Q2, Q10, and Q11, which signifies that the application is practical, easy to use, and reduces classification time and risks imposed on the deminers. Meanwhile, it is noted that participants agreed less with Q13, in which suggestions were offered to extend the capabilities of the proposed application. In general, end-users responded positively to the application of AR in aiding deminers through the improvement of UXO classification and minimisation of the hazards imposed on minefield deminers.

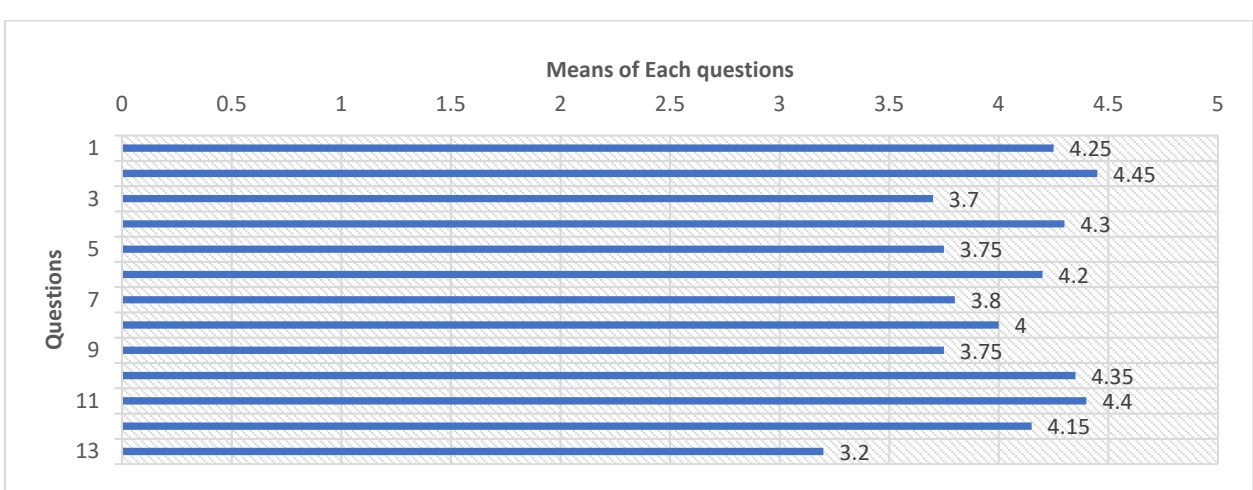

**Figure 14.** Mean of each measured item.

## 5. Limitations

Several limitations of the proposed mobile application must be addressed. Due to the infeasibility of building a UXO dataset, this application uses Vuforia object recognition for detecting and recognising UXO, which is not ideal for large datasets. In addition, the scanned object must have sufficient feature points; otherwise, the recognition will not be

successful. Deep learning methods could be introduced in the future with a UXO dataset. In addition, the simplicity of the printed UXO models must be regarded as a limitation of this study. More studies are required to test the application using real UXO in different contexts, which can be viewed as more challenging and riskier. Moreover, the small number of participants is also considered a limitation in the study. Thus, additional studies and testing on a larger number of participants must be conducted to explore various aspects of the study.

## 6. Conclusions and Future Work

This study introduced an augmented reality (AR) mobile app designed to assist minefield operators in recognising and classifying UXO and providing information on the classified UXO and how to handle it. The AR application works on-site through a mobile device and was designed and implemented in Unity 3D software with Vuforia SDK. In addition, 2D image visualisation is also rendered that shows the components that construct the UXO in a disassembled fashion. As working with real and active UXO samples was unfeasible, 3D printing technology was employed to print replicas of four UXO samples, namely vs-50, PMN-2, VS-MK, and 45 mm mortars, to build the database. These four UXO samples were selected based on their shapes and widespread use.

The application was tested to measure and ensure the soundness of the application's performance. A series of preliminary tests were performed to evaluate the application's functionality according to the following aspects: accuracy, lighting, and distance. The testing results revealed that the application could successfully perform in excellent and moderate lighting with a distance of 10 to 35 cm. However, the AR application could not function in insufficient lighting. As for the recognition accuracy, each UXO sample was scanned 20 times and the total recognition accuracy was calculated. The recognition rates of VS-50 and VS-MK were 100% and 95%, respectively. In comparison, the recognition rates of PMN-2 and 45 mm mortars were 75% and 60%, respectively. The overall recognition success rate reached 82.5%. The difference in the recognition rates due to the disparity in the number of features of each object affected the accuracy of object recognition. Furthermore, usability testing based on ISO 9421-11 standards was employed to evaluate the AR application. A questionnaire of 13 questions was conducted and submitted by 20 deminers. The usability questionnaire was based on three elements: satisfaction, efficacy, and efficiency. The results showed that the application reduced the time required to classify the unexploded ordnance object, and reduced the hazards imposed on the deminer during the demining process. Based on the survey results, we can report that AR technology is an excellent medium to aid minefield operators during the demining or recognition process, help reduce cognitive load, and minimise hazardous human errors that can put the operators in a life-threatening situation. Thus, the proposed AR application has established its capability to help deminers to complete complex tasks. Future work will explore the prospects of implementing advanced computer vision algorithms, such as deep learning, to improve object recognition and test the application in real scenarios. In addition, creating a complete UXO dataset is mandatory to increase the number of UXO objects to which such technology can be applied. In addition, utilising the Microsoft HoloLens as a display device is suggested to make the AR experience more comfortable.

**Author Contributions:** Conceptualisation, Q.A.H., H.A.H. and M.A.A.; methodology, M.A.A. and H.A.H.; software, Q.A.H., R.D.I. and M.B.O.; investigation, H.A.H., M.A.A., Q.A.H., M.M.S., R.D.I. and M.B.O.; data curation, H.A.H. and M.A.A.; writing—original draft preparation, Q.A.H., H.A.H., M.A.A., R.D.I. and M.B.O.; writing—review and editing, H.A.H. and M.A.A.; supervision, M.A.A. All authors have read and agreed to the published version of the manuscript.

**Funding:** This research received no external funding.

**Institutional Review Board Statement:** Not applicable.

**Informed Consent Statement:** Not applicable.

**Data Availability Statement:** All data were presented in the main text.

**Conflicts of Interest:** The author declares no conflict of interest.

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
