# Peer review of "UXO-AID: A New UXO Classification Application Based on Augmented Reality to Assist Deminers"

_computers, doi:10.3390/computers11080124_

Round 1
Reviewer 1 Report
This paper describes a study implementing a mobile AR system to improve efficiency of deminers in the field.
There are serious flaws with this work that need to be addressed before to be considered for publication.
This study is trying to find a solution to the problem of aiding a user detect and identify a UXO. For this the authors propose to use a mobile device's camera to detect a potential UXO from its appearance. For detection Vuforia 3D object detection is used. This is not an appropriate approach as this particular library is designed to provide the pose of the object for AR visualization not as a generic detection or identification tool. There are much better and flexible methods for object detection and identification (especially when the objects are known objects - in this case).
The purpose of the use of AR for this task is not clear. There is no purposeful task specific spatial augmentation. Instead of a simple 2D label in the 3D world, wouldn't it be easy and better to augment the image/video with a simple 2D label or drawing (or even a sound)?
The demonstrated use cases seem to be too simple (not very realistic). 3D printed ordnance is to simplistic. Aging and environmental effects need to be taken into account.
The usability study needs to be more rigorous. It is not clear what is being measured and against what. There is not baseline method to compare against. All the metrics measured should be for a specific function. Instead, it seems that the entire system is provided to the user to make a comment. Wouldn't it be better to ask, for example, about the benefit of the detection algorithm? If so, which one and against what baseline? Similarly, if the use of AR visualization is being measured, the visualization (or the user experience) should be defined properly and then alternatives should be presented and tested against. Anecdotal feedback from the users does not substantiate the claims made.
Author Response
First of all, we would like to thank the anonymous reviewers for their excellent suggestions and comments. Without their efforts, this manuscript would not have been in the current shape and form. Based on the valuable comments of the reviewers, we have thoroughly revised the manuscript. We hope that the reviewers will find that the current revision is up to their required standards. Below we detail our rebuttal.
Reviewer 1
Issues
Issue 1:
This study is trying to find a solution to the problem of aiding a user detect and identify a UXO. For this the authors propose to use a mobile device's camera to detect a potential UXO from its appearance. For detection Vuforia 3D object detection is used. This is not an appropriate approach as this particular library is designed to provide the pose of the object for AR visualization not as a generic detection or identification tool. There are much better and flexible methods for object detection and identification (especially when the objects are known objects - in this case). ………..>
To choose the method for 3d object recognition, we conducted a systematic literature review that focused on Augmented reality and object recognition; the paper title is (Taxonomy, Open Challenges, Motivations, and Recommendations in Augmented reality based on object recognition: Systematic Review) found in the link https://scholar.google.com/scholar?q=Taxonomy,+Open+Challenges,+Motivations,+and+Recommendations+in+Augmented+reality+based+on+object+recognition:+Systematic+Review&hl=en&as_sdt=0&as_vis=1&oi=scholart
In one of the aspects of the systematic review, we highlighted the technologies used to build an AR mobile application. We found the following research papers that used Vuforia 3d object recognition:
1- Remote interactive collaboration in facilities management using BIM-based mixed reality
2- Training System for Hybrid Vehicles Through Augmented Reality
3- DinofelisAR Demo Augmented Reality Based on Natural Features
4- Augmented Reality Performance in Detecting Hardware Components Using Marker Based Tracking Method.
The other approaches used deep learning, which requires a dataset to accomplish good results, and since there are no available UXO datasets, the deep learning approach was not suitable. However, building a UXO dataset and applying deep learning as an object recognition method is part of our future work, as mentioned on page 20 line (576- 579)…..>
“Future work will explore the prospects of implementing advanced computer vision algorithms such as deep learning to improve object recognition. In addition, creating a complete UXO dataset is mandatory to extend the number of UXO objects”
Issue 2:
The purpose of the use of AR for this task is not clear. There is no purposeful task specific spatial augmentation. Instead of a simple 2D label in the 3D world, wouldn't it be easy and better to augment the image/video with a simple 2D label or drawing (or even a sound)?.........>
We would like to clarify this point by stating that when we conducted interviews with the deminers and inquired about the types of information they needed the most during the classification process and the types of visuals they found most helpful. The deminers recommend images and 2D graphics that demonstrate the type and name of the UXO with an illustration of the UXO structure and composition to assist the new deminers. as in text on page 6 lines (247-254)…….>
“, we interviewed experts in UXO detection and clearance. Consequently, an unstructured interview with two experts working at the Explosive Control Directorate (EOD)- Salaheddin department. They shared their knowledge and experience about the UXO clearance and classification and emphasised the significance of various aspects of the UXO procedures. In addition, they provided their thoughts and suggestions to decide on the suitable approach to system designing and the appropriate aid required to support their operations in the field.”
Issue 3:
The demonstrated use cases seem to be too simple (not very realistic)..................>
Considering the danger of working with active UXO, we could not collect and work with these active UXO. We used the 3d printing technology to print four types of UXO models that were selected with the guidance of the deminers. In addition, we took the reviewer comment and a add a statement that the application need testing in real environment as in text on page 20 lines (576-579)…….>
“Future work will explore the prospects of implementing advanced computer vision algorithms such as deep learning to improve object recognition and testing the application in real scenarios”
Issue 4:
3D printed ordnance is to simplistic. Aging and environmental effects need to be taken into account………………………..>
We considered this and included many future works and recommendations regarding adopting deep learning algorithms and creating a reliable data set as mentioned on page 20 line (576- 579)…..>
“Future work will explore the prospects of implementing advanced computer vision algorithms such as deep learning to improve object recognition. In addition, creating a complete UXO dataset is mandatory to extend the number of UXO objects”
Issue 5:
The usability study needs to be more rigorous. It is not clear what is being measured and against what. There is not baseline method to compare against. All the metrics measured should be for a specific function. Instead, it seems that the entire system is provided to the user to make a comment. Wouldn't it be better to ask, for example, about the benefit of the detection algorithm? If so, which one and against what baseline? Similarly, if the use of AR visualization is being measured, the visualization (or the user experience) should be defined properly and then alternatives should be presented and tested against. Anecdotal feedback from the users does not substantiate the claims made…….>
We used the ISO 9241-11 standard that focused on how to evaluate the usability testing of application or system. As mentioned in (ISO 9241-11 Revised: What Have We Learnt About Usability Since 1998? ) that stated “ISO 9241-11 defines usability in terms of effectiveness, efficiency and satisfaction in a particular context of use. The intention was to emphasise that usability is an outcome of interaction rather than a property of a product. This is now widely accepted. However, the standard also places emphasis on usability measurement and it is now appreciated that there is more to usability evaluation than measurement.”
so, we compile questionnaire of 13 questions (5 questions for satisfaction, 4 questions for effectiveness, and 4 questions for efficiency) based on Likert scale. we used simple descriptive Statistics such as (max, min, mean) to measure each factor (satisfaction, effectiveness, efficiency) as we hope this study to be the first basic step for future studies to build more developed systems that focus on assisting deminers in their takes. Hence, we focused on the user acceptance of such applications for example the first two questions in efficiency factor focus on whether the application help reduces the time to classify the UXO and if the application reduces the danger imposed when dealing with explosive UXO.
Reviewer 2 Report
The article deals with important issues now and probably in the future.
The literature review and the introduction to the work are written correctly. I wouldn’t say I like the way the authors cite. It is probably an individual matter, but unfortunately, it does not appeal to me. I would suggest changing it. I am giving an example:
„In [7], the authors have used one-class classification to detect and locate if the buried object is UXO or clutter.”
In my opinion, it will look better this way: „K. Tbarki at all [7] ... "
The theoretical operation of the mobile software is described well, but the practical part lacks a clear explanation of the interface. In my opinion, it should be presented with equal detail. The Authors describe that the application has two different interfaces. Then they present only the main screen. This drawing is not done professionally. You can see the interface graphics pasted into the smartphone photo. It doesn't look good.
The tests were conducted on a small group, but the research procedure is correct. The only consideration I have is for the terms used. This is a problem translating from the authors' native language into English.
"In addition, scanning in lighting less than 12 LUX ..."
Lux is a unit of illuminance, not “lighting”. This should also be corrected in Table 2.
Author Response
Response to the Reviewers’ Comments
First of all, we would like to thank the anonymous reviewers for their excellent suggestions and comments. Without their efforts, this manuscript would not have been in the current shape and form. Based on the valuable comments of the reviewers, we have thoroughly revised the manuscript. We hope that the reviewers will find that the current revision is up to their required standards. Below we detail our rebuttal.
Reviewer2
Major Strengths:
The article deals with important issues now and probably in the future.
Mainer Issues
Issue 1:
The literature review and the introduction to the work are written correctly. I wouldn’t say I like the way the authors cite. It is probably an individual matter, but unfortunately, it does not appeal to me. I would suggest changing it. I am giving an example:
„In [7], the authors have used one-class classification to detect and locate if the buried object is UXO or clutter.”
In my opinion, it will look better this way: „K. Tbarki at all [7]…………………>
we took into consideration the reviewer suggestion and it became more powerful as in the text on pages 2 line 69…..>
“Recently, researchers developed artificial intelligence-based strategies to assist specialists and human operators in detecting explosives. K. Tbarki at all [7], the authors have used one-class classification to detect and locate if the buried object is UXO or clutter”
Issue 2:
The theoretical operation of the mobile software is described well, but the practical part lacks a clear explanation of the interface. In my opinion, it should be presented with equal detail. The Authors describe that the application has two different interfaces. Then they present only the main screen. This drawing is not done professionally. You can see the interface graphics pasted into the smartphone photo. It doesn't look good.........>
we took into consideration the reviewer’s suggestion and add more explanations and details about the two user interfaces. In addition, we correct the drawing of the user interfaces in figure 8 and it became more powerful as in text on page 10 lines (370-375)…….>
“The application has two different user interfaces: the home interface and AR mode; the home interface is displayed when the user starts the application. Figure 8-a shows the home interface with two buttons; open camera and exit buttons. Figure 8-b shows the second interface, AR mode, which opens the camera view of the mobile’s camera feature, allowing the users to scan
|
the target object. In addition, the interface has Screen-shot, Back, and exit buttons”.
|
Issue 3:
The tests were conducted on a small group, but the research procedure is correct. The only consideration I have is for the terms used. This is a problem translating from the authors' native language into English.................>
we took into consideration the reviewer suggestion and we will conduct a proof reading.
Issue 4:
"In addition, scanning in lighting less than 12 LUX ..."
Lux is a unit of illuminance, not “lighting”. This should also be corrected in Table 2...................>
We thank the reviewer for this commendable suggestion; thus, we took into consideration the valuable reviewer's suggestion, and we corrected the issue by using the term "light intensity" since lux (unit of illuminance) is a measurement of light intensity and the correct table 2 as in the text on pages 13 line 423……>
" “In addition, scanning in light intensity of less than 12 LUX made UXO objects undetectable and unrecognisable, preventing them from being identified."
|
Distance (cm) Light intensity |
10 |
15 |
20 |
25 |
30 |
40 |
|
2445 (LUX) |
YES |
YES |
YES |
YES |
YES |
NO |
|
2197 (LUX) |
YES |
YES |
YES |
YES |
YES |
NO |
|
1328 (LUX) |
YES |
YES |
YES |
YES |
YES |
NO |
|
843 (LUX) |
YES |
YES |
YES |
YES |
YES |
NO |
|
431 (LUX) |
YES |
YES |
YES |
YES |
NO |
NO |
|
380 (LUX) |
YES |
YES |
YES |
YES |
NO |
NO |
|
177 (LUX) |
YES |
YES |
YES |
YES |
NO |
NO |
|
35 (LUX) |
YES |
YES |
YES |
NO |
NO |
NO |
|
Less than 12 (LUX) |
NO |
NO |
NO |
NO |
NO |
NO |

Reviewer 3 Report
Presented article with Title ” UOX-AID: A new UXO classification application based on Augmented Reality to assist deminer” is writing on 22 pages with 14 figures, 9 tables and 42 references. The paper is well written and very clear in any part (introduction, methods, analysis and discusion), and I think deserves pubblication.
Suggestions:
- All figures need Legend X,Y axis
- Conclusions require redrafting. They are too short and inconsistent.
All the specific comments can be followed in reviewed copy of the manuscript.
I recomend this paper publish in journal after minor revision.
Author Response
First of all, we would like to thank the anonymous reviewers for their excellent suggestions and comments. Without their efforts, this manuscript would not have been in the current shape and form. Based on the valuable comments of the reviewers, we have thoroughly revised the manuscript. We hope that the reviewers will find that the current revision is up to their required standards. Below we detail our rebuttal.
Reviewer 3
Major Strengths:
Presented article with Title ” UOX-AID: A new UXO classification application based on Augmented Reality to assist deminer” is writing on 22 pages with 14 figures, 9 tables and 42 references. The paper is well written and very clear in any part (introduction, methods, analysis and discussion), and I think deserves publication.
Mainer Issues
Issue 1:
All figures need Legend X,Y axis …..>
|
|
we took into consideration the reviewer suggestion and add legend to the graphs and it became more powerful as in the figures 12,13, and 14 on pages 17 and 19…..>
Issue 2:
Conclusions require redrafting. They are too short and inconsistent..........>
we took into consideration the reviewer’s suggestion and add more explanations and details in the conclusion. as in text on page 20 lines (546-580)…….>
“This study introduced an augmented reality (AR) mobile app designed to assist minefield operators in recognising and classifying UXO and providing information on the classified UXO and how to handle them. The AR application work on-site through a mobile device and has been designed and implemented in Unity 3D software with the Vuforia SDK. In addition, 2D image visualisation is also rendered that shows the components that construct the UXO in a disassembled fashion. Considering working with real and active UXO samples was unfeasible, 3d printing technology was em-ployed to print a replica of four UXO samples, namely vs-50, PMN-2, VS-MK, and 45mm mortars, to build the database. These four UXO samples were selected based on their shapes and widespread.
The application has been tested to measure and ensure the soundness of the ap-plication's performance. A series of preliminary tests were performed to evaluate the application functionality according to the following aspects: accuracy, lighting, and distance. The testing result revealed that the application could successfully perform in excellent and moderate lighting with a distance of 10 to 35 cm. However, the AR ap-plication could not function in insufficient lighting. As for the recognition accuracy, each UXO sample was scanned 20 times and calculated the total recognition accuracy. The recognition rate of VS-50 and VS-MK were 100% and 95%. In comparison, the recognition rate of PMN-2 and 45mm mortars were 75% and 60%, The overall success rate of recognising reaches 82.5%. The difference in the recognition rate due to the disparity in the number of features of each object affects the accuracy of object recog-nition. Furthermore, usability testing based on ISO 9421-11 standards was employed to evaluate the AR application. A questionnaire of 13 questions has been conducted and submitted by 20 deminers. The usability questionnaire was based on three elements, satisfaction, efficacy, and efficiency. The results showed that the application reduced the time required to classify the unexploded ordnance object and reduced the hazard imposed on the deminer during the demining process. Based on the survey results, one can report that AR technology is an excellent medium to aid the minefield operators during the demining or recognition process and help reduce cognitive load and mini-mise hazardous human errors that can put the operators in a life-threatening situation. Thus, the proposed AR application has established its capability to help deminers to complete complex tasks. Future work will explore the prospects of implementing ad-vanced computer vision algorithms such as deep learning to improve object recogni-tion. In addition, creating a complete UXO dataset is mandatory to extend the number of UXO objects. Also, utilising the Microsoft HoloLens as a display device is suggested to make the AR experience more comfortable.”

Round 2
Reviewer 1 Report
The reviewers concerns on the first version of the paper have not been addressed in the revised manuscript (there are no changes in the revised paper that addresses the reviewer's concerns). The responses outside the manuscript partially explains some of the points but they do not make the manuscript any different than the first version.
Author Response
Dear reviewer
We appreciate you for your precious time reviewing our paper and providing valuable comments. The authors have carefully considered the comments and tried our best to address every one of them. Your concerns are very important, but it is not easy for the authors to address them in this paper because they may be one of the limitations of this research. Therefore, we were added to the limitations section in addition to the future work part in the conclusion. We thank you again, and we hope you will understand our way of answering your questions. With regards and appreciation.
Reviewer 2 Report
The article has been revised according to the reviewer's suggestions.
Author Response
Dear reviewer
We appreciate you for your precious time reviewing our paper and providing valuable comments.
Round 3
Reviewer 1 Report
The reviewer's concerns on the first version of the paper still have not been addressed in the final revised manuscript. The authors state that these would be future work. The manuscript in this form is not sufficient to advance the state of the art.